# Clinical and economic burden of acute otitis media caused by *Streptococcus pneumoniae* in European children, after widespread use of PCVs–A systematic literature review of published evidence

**Heloisa Ricci Conesa** [1], **Helena Skröder** [1], **Nicholas Norton** [1]*, **Goran Bencina** [2], **Eleana Tsoumani**[3]

1 Quantify Research, Stockholm, Sweden, 2 Center for Observational and Real-World Evidence, MSD, Madrid, Spain, 3 Center for Observational and Real-World Evidence, MSD, Athens, Greece

* nicholas.norton@quantifyresearch.com

## Abstract

### Background

Acute otitis media (AOM) is a common childhood disease frequently caused by *Streptococcus pneumoniae*. Pneumococcal conjugate vaccines (PCV7, PCV10, PCV13) can reduce the risk of AOM but may also shift AOM etiology and serotype distribution. The aim of this study was to review estimates from published literature of the burden of AOM in Europe after widespread use of PCVs over the past 10 years, focusing on incidence, etiology, serotype distribution and antibiotic resistance of *Streptococcus pneumoniae*, and economic burden.

### Methods

This systematic review included published literature from 31 European countries, for children aged ≤5 years, published after 2011. Searches were conducted using PubMed, Embase, Google, and three disease conference websites. Risk of bias was assessed with ISPOR-AMCP-NPC, ECOBIAS or ROBIS, depending on the type of study.

### Results

In total, 107 relevant records were identified, which revealed wide variation in study methodology and reporting, thus limiting comparisons across outcomes. No homogenous trends were identified in incidence rates across countries, or in detection of S. pneumoniae as a cause of AOM over time. There were indications of a reduction in hospitalization rates (decreases between 24.5–38.8% points, depending on country, PCV type and time since PCV introduction) and antibiotic resistance (decreases between 14–24%, depending on country), following the widespread use of PCVs over time. The last two trends imply a potential decrease in economic burden, though this was not possible to confirm with the identified

**Data Availability Statement:** Data underlying the results are in the Supporting Information files.

**Funding:** This study has been financed by Merck Sharp & Dohme (MSD). MSD was involved in reviewing the study design, data collection and analysis. In addition, MSD has been responsible for the decision to publish as well as reviewing the manuscript preparation. The funder (MSD) provided support in the form of salaries for authors ET and GB. The specific roles of these authors are articulated in the 'author contributions' section.

**Competing interests:** ET and GB are employees of MSD. HRC, HS and NN are employees of Quantify Research, which received consulting fees from MSD to support the preparation and development of the manuscript. This does not alter our adherence to PLOS ONE policies on sharing data and materials. The authors report no other conflicts of interest in this work.

cost data. There was also evidence of an increase in serotype distributions towards non-vaccine serotypes in all of the countries where non-PCV serotype data were available, as well as limited data of increased antibiotic resistance within non-vaccine serotypes.

## Conclusions

Though some factors point to a reduction in AOM burden in Europe, the burden still remains high, residual burden from uncovered serotypes is present and it is difficult to provide comprehensive, accurate and up-to-date estimates of said burden from the published literature. This could be improved by standardised methodology, reporting and wider use of surveillance systems.

## Introduction

Acute otitis media (AOM) is an inflammatory disease of the middle ear associated with middle ear effusion and symptoms like ear pain, fever, irritability, otorrhea, anorexia, and sometimes vomiting or lethargy [1, 2]. Even though AOM can also have a viral etiology, it is more commonly the result of a bacterial infection by *Streptococcus pneumoniae (S. pneumoniae)*, *Hemophilus influenzae*, or *Moraxella catarrhalis*, and is often treated with pain medications and/or antibiotics, depending on the patient's age and severity of symptoms [3, 4].

Pneumococcal conjugate vaccines (PCVs) confer protection against diseases caused by vaccine serotype *S. pneumoniae* [5]. Since *S. pneumoniae* is the most common cause of AOM, vaccination of children younger than 5 years with PCVs can reduce the risk of AOM, while further protecting the population and mitigating the spread of antibiotic-resistant serotypes [6–9].

European countries started introducing PCVs in the early 2000s, beginning with PCV7, which protected against 7 pneumococcal serotypes (4, 6B, 9V, 14, 18C, 19F, and 23F) [5, 10]. Previous literature has posited that widespread use of PCVs in higher-income countries has shifted both AOM etiology and pneumococcal serotype distribution among the target populations [11]. The ongoing changes in etiology and emergence of non-vaccine serotypes are a reason for growing concern in the medical community around antibiotic-resistant bacteria as a major cause of treatment failure in pediatric patients with AOM [12, 13]. Furthermore, a growing body of literature has focused on the emergence of residual burden, as serotypes of *S. Pneumoniae* not covered by available PCVs continue to circulate in the pediatric population [14–18]. As a consequence, higher-valent vaccines have been introduced in immunization programs, such as PCV10 (with additional serotypes 1, 5, and 7F) and PCV13 (with additional serotypes 3, 6A, and 19A) [5]. S1 Fig in the appendix provides an overview of which PCVs were introduced in each of the European countries of interest between 2002 and 2021 [9, 19–28]; by 2011, 70% of the included countries had introduced a PCV into their national immunization program (NIP) for children.

Recently, two higher valency PCVs have been approved by the European Medicines Agency for adult use and are soon expected to have pediatric indication; in addition to the serotypes present in previous PCVs, PCV15 contains 22F and 33F, and PCV20 contains serotypes 8, 10A, 11A, 12F, and 15B [5, 29–31].

Since AOM is one of the most common pediatric infectious diseases, its burden is substantial [4]. A 2012 estimate of the global yearly incidence rate of AOM is 108.5 cases per 1000 person years, with more than 700 million cases estimated to occur every year and 51% of these

cases occurring in children [32]. In central Europe, the average incidence rate is 3.6% per year [32]. Incidence rates are highest in children 0–4 years of age, with a peak in the first year of life [32]. In addition, AOM is both a leading cause of antibiotic prescriptions and a major contributor of medical costs in children [4].

However, the incidence and prevalence of AOM, serotype distribution of the AOM-causing bacteria, *S. pneumoniae*, and the burden of the disease vary between countries and over time. Furthermore, previous reviews of published literature may lack current and comprehensive evidence for a number of reasons: they may no longer be up to date, may not look at all relevant metrics for burden or may be limited in number of countries included [4, 33–35]. Therefore, the objectives of this review were to identify available evidence in published literature related to the burden of AOM over the past 10 years, for 31 European countries, based on the following factors: incidence, etiology, changes in serotype distribution and antibiotic resistance over time for *S. pneumoniae*, costs and impact on quality of life. The authors then aimed to provide comparable estimates, where possible, that quantified this burden for children <5 years of age around Europe within the context of increased PCV usage.

## Methods

This systematic literature review was carried out using methods based on the guidelines by PRISMA, the Preferred Reporting Items for Systematic Reviews and Meta-Analyses [36]. The study protocol for this project was registered with PROSPERO, the international prospective register of systematic reviews on 19-Dec-2021. The registration ID is CRD42021292105, and it can be accessed at: https://www.crd.york.ac.uk/prospero/display_record.php?RecordID=292105.

### Eligibility criteria

To be included in this review, records had to fulfil the inclusion criteria described in Table 1.

### Information sources and search strategy

Searches were conducted in PubMed® and Embase®. The search terms (S1 Table) were divided into six blocks (population, exposure, epidemiology and etiology, burden, *S. pneumoniae* and VCR). Each block included synonyms and MeSH-terms and were searched for in different combinations, also filtering by English language and date of publication (after 1$^{st}$ Jan 2011). As a complement to the database searches, one ad-hoc Google search was performed per country (countries listed in Table 1) using standardized Google searches. In addition, the websites of three relevant conferences were used to search for abstracts: the European Congress of Clinical Microbiology & Infectious Diseases, the European Society for Pediatric Infectious Diseases, and the European Academy of Pediatrics [37–39]. All searches were conducted between 2021, November 11$^{th}$ and 12$^{th}$.

### Selection process

After removal of duplicate records, a two-step process was employed for the selection process. In step one, the titles and abstracts of articles identified in PubMed®, Embase®, Google or the conference websites were assessed and categorized as 'included,' 'unsure,' or 'rejected' independently by two reviewers, based on the eligibility criteria. Discrepancies between reviewers were resolved by consensus; unresolved disputes were referred to a third arbitrator and a consensus reached. In step two, the two reviewers obtained and independently reviewed the full text records in the 'included' and 'unsure' categories. Reasons for exclusions of records

**Table 1. Inclusion criteria.**

|  | Inclusion Criteria |
|---|---|
| **Population(s)** | Children younger than 5 years of age (when specified, children up to 5 years and 11 months were included) in European countries (Austria, Belgium, Bulgaria, Croatia, Cyprus, Czechia, Denmark, Estonia, Finland, France, Germany, Greece, Hungary, Iceland, Ireland, Italy, Latvia, Lithuania, Luxembourg, Malta, Netherlands, Norway, Poland, Portugal, Romania, Slovakia, Slovenia, Spain, Sweden, Switzerland, United Kingdom). |
| **Intervention/Exposure** | Acute otitis media[1] (AOM), or at risk of developing AOM |
| **Comparisons** | Not applicable |
| **Outcomes (contain data for any of the following)** | • Epidemiological data on AOM's:<br>  ○ Incidence<br>  ○ Prevalence<br>  ○ Etiology<br>• Serotype data on *S. pneumoniae*<br>  ○ Serotype distribution of *S. pneumoniae*<br>  ○ Antibiotic resistance and susceptibility of *S. pneumoniae*<br>• Economic data on AOM's:<br>  ○ Resource use<br>  ○ Costs<br>• Impact on QoL (humanistic burden)<br>• Vaccine coverage rates (VCR) of PCVs[2] |
| **Time** | Records published since 01 January 2011[3]. |
|  | Conference abstracts published from January 1st, 2017 to the latest available year, as of November 12th, 2021. |
| **Study design** | • Observational studies (for instance cross-sectional, cohort or case-control)<br>• Economic studies (for instance cost of illness)<br>• Systematic reviews |
| **Other** | For all records, the following should apply:<br>  • Available in full text, unless the material is a conference abstract<br>  • Available in English |

1. To avoid excluding records that refer to AOM in the full text but do not provide details in the title and abstract, the term "otitis media" was included in the search strategy, but records that did not contain evidence of AOM were excluded.

2. Outcomes on VCR of PCVs were excluded from the present study.

3. **Records published after 01 Jan 2011 might include information collected before 2011. The time period from when the data was collected was extracted as a separate data element in order to differentiate it from the publication date.

were recorded. Any disagreements were referred to the third arbitrator. A PRISMA flow diagram (Fig 1) was used to visualize the number of records included and excluded at each stage of the review. Records excluded at the full paper stage were tabulated alongside the reason for exclusion in accordance with best practice guidelines [36].

## Data collection process

All relevant data from the included records was extracted by one reviewer and a quality check was performed by a second reviewer, comparing the extracted content of 10% of the included records to the original work, for correctness. Any major discrepancies resulted in a further review of 5% of the remaining records, and so forth, until free of discrepancies. The type of data extracted is presented in S2 Table. For each of the outcome variables, any stratification by age group, level of complication, etc. was recorded separately if available.

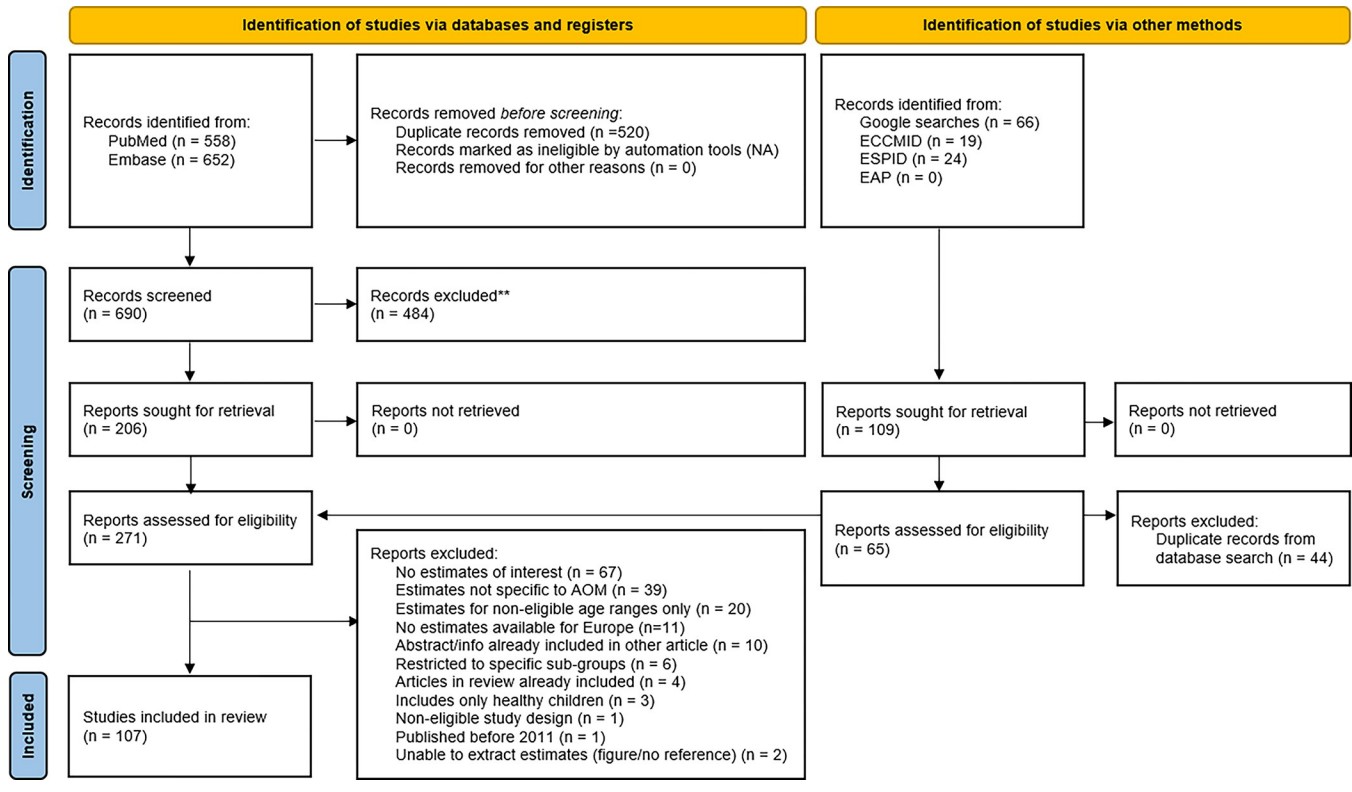

**Fig 1. PRISMA flow diagram for the literature search.**

## Risk of bias assessment

As different study designs were included in the review, three tools were selected to assess risk of bias for the respective record types. For observational studies and health economic studies that do not involve modelling (for instance, studies assessing economic burden), the ISPOR-AMCP-NPC was chosen due to its applicability for multiple study designs [40]. For health economic studies involving modelling, the ECOBIAS risk of bias tool for model-based economic evaluations was chosen [17]. For systematic reviews, ROBIS was employed since it is a rigorous tool that was specifically designed to assess risk of bias in this type of study [41, 42]. A single reviewer assessed the risk of bias. During this process, the reviewer made a list of the included records that were found to be most difficult to assess. These records were then assessed by a second reviewer and discussed together with the original reviewer to reach a resolution. Unresolved assessments were referred to a senior arbitrator and a consensus reached. There is a potential risk for publication bias in all types of published literature. In order to address this for the context of this study, the reviewers looked for conflicts of interest reported by the authors in each record, as well as their sources of funding to assess whether there could be a potential risk. More advanced methods were not applicable in this review, as the records included are not homogenous in study design. After the risk of bias assessment, each record was categorized as either "high," "medium" or "low" risk of bias. The results of the assessment are provided in the list of included records, provided in S3 Table.

## Synthesis of results

The findings for each of the objectives were grouped into the following objective categories: epidemiology, antibiotic resistance, serotype data, economic burden and quality of life. Data

on the outcomes reported in each of the included studies were collected and stratified by country, population and year. As earlier estimates of the burden are available in previously-published literature [4, 33–35], the authors chose to focus on estimates published between 2011 and 2021 (the last 10 years).

Changes in AOM burden were analyzed by comparing estimates across the study period, taking into account the introduction of newer PCVs. Estimates for select metrics (e.g., incidence, direct medical costs, antibiotics reviewed, etc.) were presented together in tables and/or figures, where the comparison of estimates between countries was deemed feasible. For the purpose of comparability, age groups originally reported in years were transformed into months and time intervals reported in months were rounded up to years.

The gaps in the identified data were assessed, and possible relationships in the data were explored and described, where possible. Since no statistical analysis was conducted, absence of data (for instance, from a certain geographical area) was addressed through discussion only. Though records and outcomes related to vaccine coverage rates were included in the scope of the review, the synthesis of these results was not reported in the present study due to risk of bias further described in the Discussion. Nevertheless, the records pertaining to PCV-coverage were still included in the summary statistics as they were indeed part of the original search and extraction process.

## Results

### Study selection and general characteristics

In total, the searches generated 690 unique records from PubMed® and Embase®. In addition, 109 records were sought for retrieval after the ad hoc Google search and the conference abstract website searches. After title and abstract review of all collected records, a total of 271 records were kept for full-text review. Finally, 107 records were included after the full-text review (see the PRISMA flow chart presented in Fig 1 and the PRISMA checklist in S8 Table). The full list of included records, including the risk of bias assessment result, is available in S3 Table. The countries with the highest number of included records were Sweden, Spain, Italy, Germany and France, while no available data were identified for 6 countries (Austria, Czech Republic, Ireland, Latvia, Luxembourg and Malta) (Fig 2).

Many of the included studies had information regarding several of the outcomes of interest. The main proportion of the included studies (69%) had information regarding the epidemiology of AOM, while *S. pneumoniae* information (serotype distribution and antibiotic resistance) was available in approximately 36%, and economic data in 34%. Even though there was a higher number of papers in the economic outcome category, only 19 of them included QoL-data. Details on which countries had data in each respective area can be found in S4 Table.

The included studies were rather evenly distributed over the inclusion years (2011–2021), with a peak of epidemiology papers between 2017 and 2019 (Fig 3). For the data synthesis, 78 studies (73%) reported data collected between 2011 to 2021, covering 21 countries.

### Epidemiology

**Incidence of AOM.** Of the 61 studies that reported data on incidence rates of AOM, 12 contained information from the period 2011–2021, providing recent estimates for 11 of the 31 included countries (Table 2). One study, Chapman et al 2020, assumed the same incidence rate for several countries, and was therefore excluded from the synthesis of epidemiological data [43]. When needed, incidence rates were transformed to a rate of 100,000 person-years. Incidence rates of AOM ranged from 630 cases (in children aged 3 to <7 years in 2011–2013)

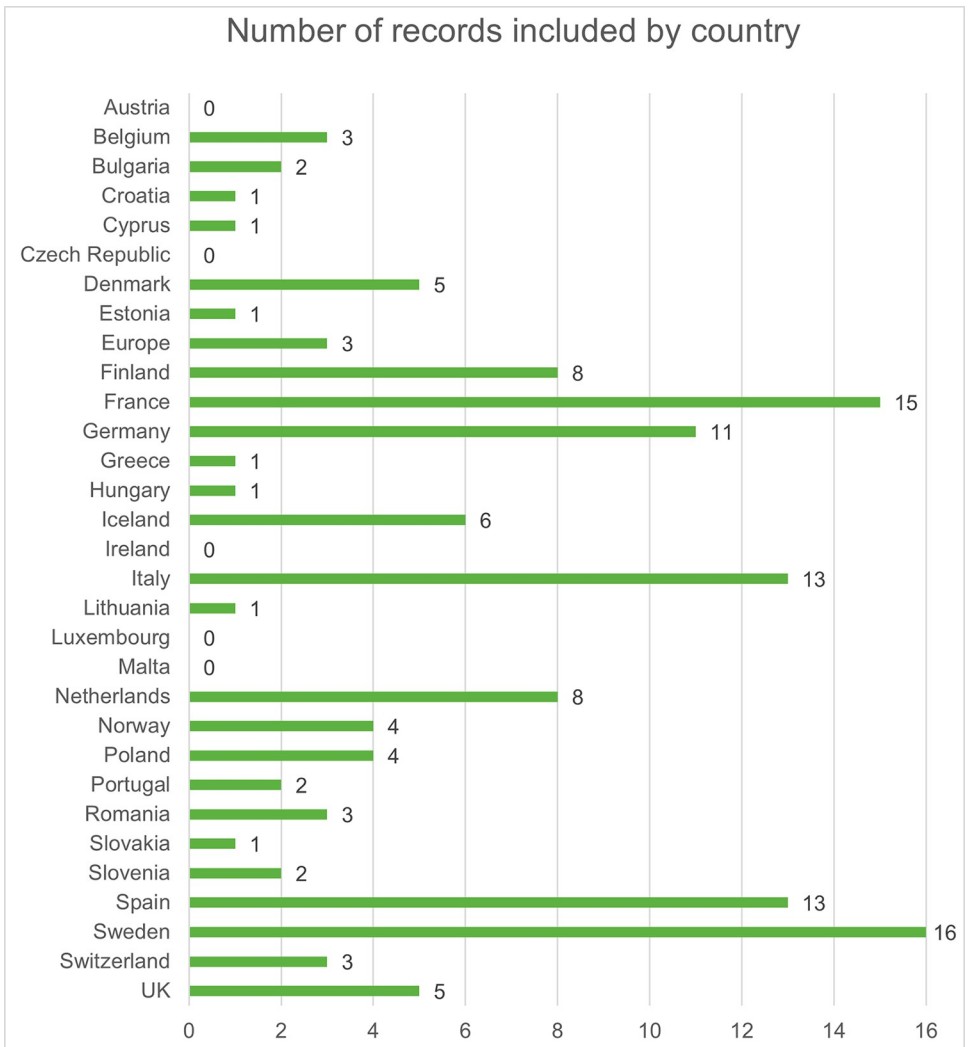

**Fig 2. The number of studies with data for each of the included countries.** Note: some of the included papers reported on more than one outcome.

to 62,400 cases (in children aged 6 to 12 months in 2008–2014) per 100,000 person-years [44, 45].

As presented in Table 2, incidence estimates in across time were available for 4 countries. (Germany, Iceland, Italy and Sweden). In Germany, the incidence of AOM was estimated to decrease over a 5-year period (2012 to 2017), during which PCV10 and PCV13 were part of the NIP. This decrease was seen in both groups of children younger than 24 months (from 3,500 to 2,700 cases per 100,000 person-years) and between 25–59 months (from 4,100 to 3,400 cases per 100,000 person-years) [48]. A similar decrease was estimated in two separate studies in Iceland after the introduction of PCV10. The methodologies for estimating the incidence differed in the two studies; higher rates were found in the study estimating incidence of AOM from diagnoses in primary care than when assessing middle-ear fluid (MEF) samples of children with ruptured tympanic membranes or middle-ear inflammation [45, 49]. After the introduction of PCVs in three Swedish regions (one introduced PCV10, another PCV13 and the third started with PCV13 and replaced it with PCV10) a decrease in AOM incidence was

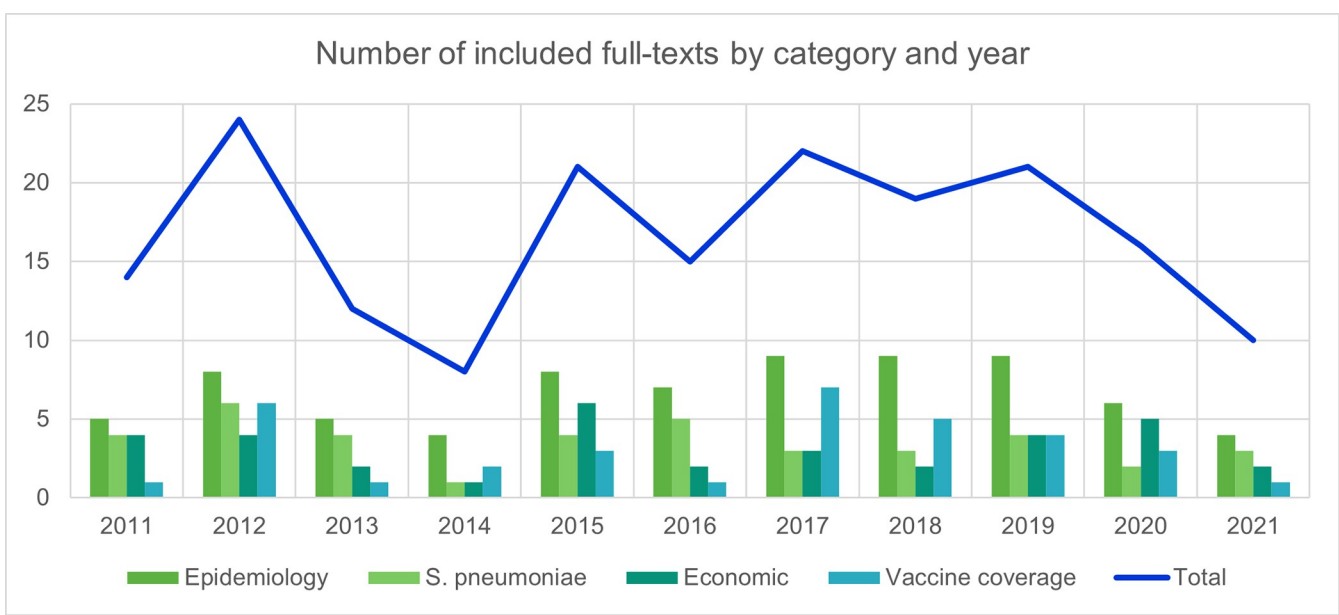

**Fig 3. Number of papers included by year and outcome category.** Note: some of the included papers reported on more than one outcome.

seen in all three regions, in all age groups with the exception of 0–35 months in Skåne [53]. Contrary to the studies from the other three countries, estimates from the Italian region of Veneto showed a slightly-increasing incidence rate of AOM among children 24–59 months (11,900 to 12,900 cases per 100,000 person-years) when PCV13 was introduced in 2010 compared to 2017 [50, 54]. With limited trend data over time it is difficult to draw conclusions

**Table 2. Incidence rates of AOM cases.**

| Location | Population | Data collection period | PCV used in NIP during the period | Incidence rate (per 100,000 person-years) |
|---|---|---|---|---|
| Croatia [46] | 1–59 months | 2012 | None | 23,581 |
| Estonia [47] | 3–71 months | 2011–2012 | None | 13,780 |
| Germany [48] | <24 months | 2012 | PCV10/PCV13 | 3,500 |
| | | 2017 | PCV10/PCV13 | 2,700 |
| | 24–59 months | 2012 | PCV10/PCV13 | 4,100 |
| | | 2017 | PCV10/PCV13 | 3,400 |
| Iceland [44] | <12 months | 2011–2013 | PCV13 | 9,800 |
| | 12 –<24 months | | | 10,580 |
| | 24 –<36 months | | | 3,150 |
| | 36 –<84 months | | | 630 |
| Iceland [45] | <12 months | 2008–2013 | None→PCV7→ PCV10 | 48,300 |
| | | 2014 | PCV10 | 34,200 |
| | 12 to <24 months | 2008–2013 | None→PCV7→ PCV10 | 57,200 |
| | | 2014 | PCV10 | 51,900 |
| | 24 to <36 months | 2008–2013 | None→PCV7→ PCV10 | 29,100 |
| | | 2014 | PCV10 | 25,500 |
| | < 36 months | 2005–2015 | None→PCV7→ PCV10 | 41,700 |
| | | 2008–2013 | None→PCV7→ PCV10 | 43,600 |
| | | 2014 | PCV10 | 38,000 |

*(Continued)*

**Table 2.** (Continued)

| Location | Population | Data collection period | PCV used in NIP during the period | Incidence rate (per 100,000 person-years) |
|---|---|---|---|---|
| Iceland [49] | 0–84 months | 2012–2014 | PCV10 | 2,350 |
| | | 2015–2017 | PCV10 | 1,330 |
| Italy, Veneto Region [50] | 24–59 months | 2010 | PCV13 | 11,900 |
| | | 2017 | PCV13 | 12,900 |
| Lithuania [47] | 0–71 months | 2011–2012 | None | 18,400 |
| Netherlands, Utrecht [51] | 0–23 months | 2007–2012 | PCV7→ PCV10 | 56,900 |
| Netherlands [52] | 6–12 months | 2008–2014 | PCV7→ PCV10 | 62,400 |
| Poland [47] | 0–71 months | 2011–2012 | PCV7 (risk groups) | 11,570 |
| Romania [47] | 2–71 months | 2011–2012 | None | 14,190 |
| Slovenia [47] | 1–70 months | 2011–2012 | PCV7 (risk groups) | 34,030 |
| Sweden, Skåne Region [53] | 0–35 months | 2011 | PCV10/PCV13 | 30,544 |
| | | 2012 | PCV10/PCV13 | 37,124 |
| | | 2013 | PCV10/PCV13 | 39,868 |
| | | 2010–2013 | PCV10/PCV13 | 23,697 |
| | 36–71 months | 2011 | PCV10/PCV13 | 20,109 |
| | | 2012 | PCV10/PCV13 | 19,491 |
| | | 2013 | PCV10/PCV13 | 16,454 |
| | | 2010–2013 | PCV10/PCV13 | 17,994 |
| | ≤ 71 months | 2011 | PCV10/PCV13 | 24,837 |
| | | 2012 | PCV10/PCV13 | 23,918 |
| | | 2013 | PCV10/PCV13 | 20,207 |
| | | 2010–2013 | PCV10/PCV13 | 23,640 |
| Sweden, Västra Götaland Region [53] | 0–24 months | 2011 | PCV10/PCV13 | 31,800 |
| | | 2012 | PCV10/PCV13 | 29,432 |
| | | 2013 | PCV10/PCV13 | 25,914 |
| | | 2010–2013 | PCV10/PCV13 | 25,710 |
| | 25–60 months | 2011 | PCV10/PCV13 | 21,320 |
| | | 2012 | PCV10/PCV13 | 19,143 |
| | | 2013 | PCV10/PCV13 | 16,317 |
| | | 2010–2013 | PCV10/PCV13 | 15,270 |
| | 0–60 months | 2011 | PCV10/PCV13 | 25,671 |
| | | 2012 | PCV10/PCV13 | 24,227 |
| | | 2013 | PCV10/PCV13 | 21,411 |
| | | 2010–2013 | PCV10/PCV13 | 25,268 |
| Sweden, Västerbotten Region [54] | 0–59 months | 2011 | PCV10/PCV13 | 17,700 |
| | | 2012 | PCV10/PCV13 | 17,500 |
| | | 2013 | PCV10/PCV13 | 16,200 |
| | | 2014 | PCV10/PCV13 | 16,100 |

about the impact of PCVs on AOM incidence. The available data implied that the trends in AOM incidence differ between countries (and age groups) after widespread PCV use, with most of the studies estimating a decrease over time.

**Hospitalization rates for AOM patients.** There were two countries with studies that reported the incidence of hospitalization (i.e., inpatient stay) due to AOM between 2011 and 2021 (Table 3). Three studies (one in Iceland and two in Italy) estimated incidence rates of hospitalization before and after the introduction of PCVs (Table 3). All studies found a

**Table 3. Time trends of changes in AOM hospitalization incidence rates.**

| Location | Pop. | Pre-PCV | | Post-PCV | | Decrease in incidence rate |
|---|---|---|---|---|---|---|
| | | Time interval | Incidence (per 100,000) | Time interval | Incidence rate (per 100,000) | |
| Iceland [26] | 0–47 months | 2005–2013 (pre PCV10) | 232 | 2011–2018 (post PCV10) | 145 | 38% |
| Italy, Apulia region [55] | <60 months | 2001–2005 (pre PCV7) | 205 | 2006–2011 (post PCV7) | 154.7 | 25% |
| Italy (8 first regions to introduce PCVs) [56] | | | 91 | | 58.9 | 35% |
| Italy [56] | | | 100 | | 61.3 | 39% |

decrease of between 24.5% and 38.8% in AOM-related hospitalizations after PCVs were introduced [26, 55, 56], suggesting a reduction in either AOM severity or AOM cases over time.

**Etiology.** Etiology information was identified in 36 studies. Out of those, 21 contained data from 11 countries on *S. pneumoniae* detection in AOM patients during the publication period (2011–2021), as shown in Table 4. The remaining 15 studies presented information on detection of *S. pneumoniae* from before 2011.

The estimates in Table 4 are divided by the source of the sample: either MEF or nasopharynx (NP). Overall, *S. pneumoniae* was less present in MEF than in NP samples. In MEF samples, the presence of *S. pneumoniae* ranged from 5.8% in Germany (after the introduction of PCV10/PCV13 to their NIP) and to 70.3% in Romania (which did not include PCVs in their NIP at the time). Estimates from NP samples ranged from 10.5% in Sweden (post-PCV10/PCV13) to 73.4% in Iceland (post-PCV10) [57, 58]. Etiology data sourced from both MEF and NP swabs in the same age groups were available for three countries (France, Germany and Portugal). In all three cases, the estimates varied between sampling methods but were also lower in MEF than in NP samples. For the countries with data across time for the same age group and sampling method, only one country showed a discernable difference in the share of cases caused by *S. Pneumoniae*; there was about a 50% difference in *S. Pneumoniae* detection between two studies estimating the etiology of AOM in central Romania (70.3% to 32.9%). However, this difference could in part be explained by differences in the study populations. One study focused on AOM patients with otorrhea or who underwent tympanocentesis, while the other only focused on any presenting AOM cases.

## Antibiotic resistance

In total, there were 23 records that included data related to at least one antibiotic (S5 Table). Of these, seven studies had additional information regarding serotype-specific resistance to any of the antibiotics. Most studies were conducted in France, Spain and Belgium.

The included studies had estimates for either resistance, intermediate resistance or non-susceptibility (resistance + intermediate resistance combined) for a total of 19 different antibiotics (S5 Table). The most commonly tested antibiotic was penicillin, which was included in 96% of the records, followed by erythromycin (57%), multiple drugs (multidrug resistance, 39%), tetracycline (17%) and ceftriaxone (17%). Nearly all studies assessing penicillin susceptibility used MEF samples; there was only one study conducted in Belgium (Ekinci et al., 2021) that used NP samples [78].

In general, the estimates from the included studies revealed a decrease in the level of penicillin non-susceptibility over time, in countries or periods where PCVs had been introduced (Fig 4). The study reporting on Romania (where no PCV had been introduced into the NIP as of 2021) showed high non-susceptibility (94%) remaining consistent over the time span.

**Table 4. Detection of S. pneumoniae in children younger than 5 years with AOM in Europe between 2011-2021(% and sample size).**

| Location | Population | Period of data collection | PCV used in NIP during the period | Estimate (%, CI)* | Sample size |
|---|---|---|---|---|---|
| **Middle ear fluid** | | | | | |
| Finland [59–62] | 0–16 years | 2003–2012 | No PCV → PCV10 | 38.0% | 56 |
| | <24 months | | No PCV → PCV10 | 43.0% | 14 |
| | 6–39 months | 2010–2011 | PCV10 | 34.9% | 43 |
| | 5–42 months | | PCV10 | 31.1% | 90 |
| France [63, 64] | 6–24 months | 2011–2014 | PCV13 | 20.7% | 56 |
| | <36 months | 2015–2017 | PCV13 | 17.1% | 175 |
| Germany [57] | 2–71 months | 2011 | PCV10/PCV13 | 5.8% | 209 |
| | | 2012 | PCV10/PCV13 | 6.9% | 412 |
| | | 2013 | PCV10/PCV13 | 6.3% | 340 |
| | | 2014 | PCV10/PCV13 | 6.9% | 251 |
| | | 2015 | PCV10/PCV13 | 6.7% | 214 |
| Italy, Milan [65] | 6–96 months | 2015–2016 | PCV13 | 27.1% | 48 |
| Poland [66, 67] | 12–35 months | 2010–2014 | PCV7 (R) | 30.0% | 20 |
| | 36–71 months | | PCV7 (R) | 32.3% | 31 |
| | <60 months | 2010–2016 | PCV7 (R) | 50.0% | 407 |
| Romania [68, 69] | <60 months | 2009–2011 | No PCV | 70.3% | 111 |
| | <60 months | 2009–2014 | No PCV | 32.9% | 391 |
| Spain [70] | 3–36 months | 2009–2012 | PCV7 (P) → PCV10/PCV13 (P) | 39.3% | 117 |
| Portugal** [58] | 24–83 months | 2014–2016 | No PCV → PCV13 | 47.0% | 151 |
| **Nasopharynx** | | | | | |
| Belgium [71] | 6–30 months | 2016 | PCV13 | 69.2% | 39 |
| | | 2017 | PCV13 | 64.8%, (55.5%-73.0%) | 122 |
| France [72–75] | < 24 months | 2006–2017 | PCV7 →PCV13 | 55.9% | 9,957 |
| | 6–24 months | 2011–2012 | PCV13 | 54.5% | 1,790 |
| | | 2013–2016 | PCV13 | 54.6% | 3,649 |
| | | 2014 | PCV13 | 56.2% | 7,991 |
| | 6–35 months | 2011–2018 | PCV13 | 60.8% | 3,964 |
| Germany [57] | 2–71 months | 2011 | PCV10/PCV13 | 53.1% | 192 |
| | | 2012 | PCV10/PCV13 | 48.8% | 389 |
| | | 2013 | PCV10/PCV13 | 45.8% | 323 |
| | | 2014 | PCV10/PCV13 | 51.5% | 235 |
| | | 2015 | PCV10/PCV13 | 53.6% | 196 |
| Iceland [76] | 12–83 months | 2011 | PCV10 | 65.0% | 420 |
| | | 2012 | PCV10 | 61.1% | 465 |
| | | 2013 | PCV10 | 70.9% | 471 |
| | | 2014 | PCV10 | 69.6% | 566 |
| | | 2015 | PCV10 | 73.4% | 533 |
| | | 2016 | PCV10 | 65.6% | 541 |
| | | 2017 | PCV10 | 53.8% | 506 |
| | | 2009–2011 | PCV7 (R) →PCV10 | 71.8% | 1,380 |
| | | 2012–2017 | PCV10 | 65.9% | 3,081 |
| Portugal [58] | 24–83 months | 2014–2016 | PCV13 | 47.0% | 151 |
| Sweden, Skåne Region [77] | 0–35 months | 2017–2018 | PCV10/PCV13 | 16.1% | 684 |
| | 36–83 months | | PCV10/PCV13 | 10.5% | 191 |
| | **Nasopharynx and oropharynx** | | | | |

(*Continued*)

**Table 4.** (*Continued*)

| Location | Population | Period of data collection | PCV used in NIP during the period | Estimate (%, CI)* | Sample size |
|---|---|---|---|---|---|
| Poland [67] | 12–35 months | 2010–2014 | PCV7 (R) | 25.0% | 20 |
| | 36–71 months | | PCV7 (R) | 35.5% | 31 |

Abbreviation: CI, confidence interval. (R) Only introduced to NIP for certain risk groups. (P) Only introduced to NIP in certain regions

Notes

* Of the records included, the authors identified only one instance where the study reported confidence intervals with their Etiology estimate

** Otorrhea samples, not specified as middle ear fluid

Studies from two countries, Finland and Spain, estimated an upward trend between two periods, based on smaller sample sizes (n<40).

In studies where specific serotypes were assessed, the reported serotypes included 14, 15A, 19A, 19F, 23B, 23F, 6A, 6B, 6C, others (not specified), PCV7-specific serotypes, and serotypes other than 19A. Fig 5 presents the available data for serotype-specific non-susceptibility. Two separate studies estimated that non-susceptibility was especially high in Romania (between 50–100% for the included serotypes), where PCVs were not included in the NIP as of 2021 [68, 69]. The non-susceptibility for many serotypes appeared to be lower in the later PCV-periods compared to the earlier periods, although there were variations between countries and no

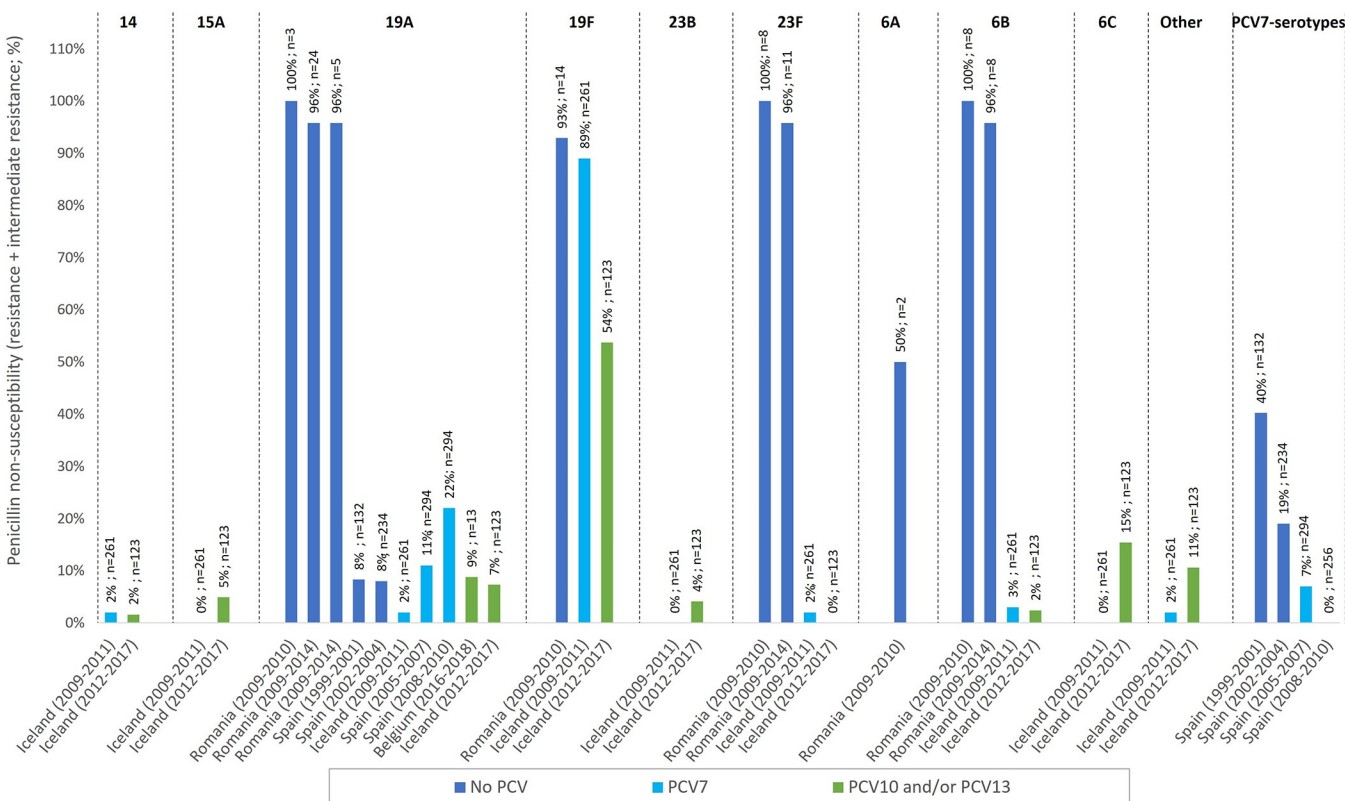

**Fig 5. Serotype-specific penicillin non-susceptibility during different PCV periods in Iceland, Romania, Spain and Belgium [68, 69, 76, 78, 84, 85].** Notes: 1. No-PCV, early PCV, later PCV.

clear trend was evident. However, when looking at estimates for serotypes not included in PCV13 (or the previous PCVs), one study in Iceland estimated slight increases in non-susceptibility (between 4%-15% over 8 years). This increase in non-susceptibility for non-vaccine serotypes has been documented in other studies [87–90].

### Serotype distribution

A total of 32 records included data related to serotype distribution. Due to the variability of study populations, sample sizes, serotypes and time span in data collection (25 years, from 1993 to 2018), it was deemed more appropriate to provide a summary of the serotype distributions reported since 2011, by country, for all serotypes covered by PCV20 or lower (Table 5). In total, 17 records contained data collected from 2011 or later, spanning 10 countries. For seven of the records, samples were taken from middle ear fluid (MEF), while 10 sampled serotypes using nasopharyngeal swabbing and one study did not specify the kind(s) of samples taken. As is evident in the table, the gaps in serotype information for countries with available data are considerable, depending on the country. No trends were discernable across countries or studies in the collected data for serotypes covered by existing PCVs.

There were also seven studies, covering five countries, which reported serotype distribution data across time for 'non-vaccine serotypes,' referring to serotypes not covered by PCV13 or lower-valency vaccines (e.g., 15A, 6C, 21, etc.). As is shown in Table 6, all seven publications reported evidence of increases in the percentage of AOM cases attributed to non-vaccine serotypes of *S. pneumoniae*. This evidence suggests an increase in AOM cases involving non-vaccine serotypes across Europe, which would be an indication of increased residual burden after the introduction of PCVs in the included countries.

### Economic burden

In total, 32 studies were identified which contained data related to the economic burden of AOM. The studies provided data for 21 countries, with some studies providing data for multiple countries. S6 Table provides summary data for those countries, including the types of cost data (direct medical, direct non-medical, indirect costs) and which studies involved economic modelling. Direct medical cost, direct non-medical cost and indirect cost estimates were available in 91%, 55% and 82% of countries, respectively.

Direct medical costs were the most common economic burden outcome represented in the data. The total direct medical costs estimated by the included records varied by country and record, ranging from €10.98 in the UK to €284.35 in Sweden. Total costs (including direct medical, direct non-medical and indirect costs) were reported for nine countries, across four studies [35, 96–98]. The latest estimates for each country ranged from €170 in the Germany to €693 in Sweden [35, 96, 97]. The most recent estimates of direct medical costs and total costs from each available country are presented together in Fig 6 [35, 96–100]. With the exception of the Netherlands, the variation between countries in the latest estimates for direct and total costs mirrored each other. However, due to variations in the methods and costs included by each study, no conclusions could be drawn from the available evidence.

### Quality of life (humanistic burden)

Nineteen studies provided QoL data for 19 European countries. Out of those, 10 use the same utility decrement of 0.005 QALY per AOM episode, which was obtained from Oh et al, 1996 and Melegaro and Edmunds, 2004 [101, 102]. Furthermore, one study considered a disutility of 0.09 QALYs for hearing loss due to AOM while two other studies used a disability weight of 0.02 DALYs [46, 103]. The list of studies that report QoL data is presented in S6 Table.

**Table 5. Frequency (%) of PCV-serotypes identified between 2011–2021 by testing source and country.**

| Location | Data period | PCVs included during period | Population (n, by order of publication) | PCV7 | | | | | | | | PCV10 | | | PCV13 | | | | | | PCV15 | | PCV20 | | | | |
|---|---|---|---|---|---|---|---|---|---|---|---|---|---|---|---|---|---|---|---|---|---|---|---|---|---|---|---|
| | | | | 4 | 6B | 9V | 14 | 18C | 19F | 23F | PCV7 serotypes | 1 | 5 | 7F | 3 | 6A | 19A | 1, 3, 5, 6A, 7F, 19A | 3, 6A, 19A | PCV13 serotypes | 22F | 33F | 8 | 10A | 11A | 12F | 15B |
| **Middle ear fluid** | | | | | | | | | | | | | | | | | | | | | | | | | | | |
| France [11] | 2011 | 13 | 0–16 years (n = 152) | 0% | 0% | 0% | 1% | 1% | 10% | 1% | 13% | | | | | | 38% | 49% | | | | | | | | | |
| Germany [57] | 2011 | 10/13 | 2–71 months (n = 12) | | | | | | 8% | | | | | 8% | 33% | | 33% | | | | | | 8% | | | |
| | 2012 | 10/13 | 2–71 months (n = 28) | | | | | | 4% | | | 4% | | | 32% | | 21% | | | | | 4% | 4% | 18% | | |
| | 2013 | 10/13 | 2–71 months (n = 21) | | | | | | 5% | 5% | | | | | 38% | | 10% | | | | 5% | | | 5% | | |
| | 2014 | 10/13 | 2–71 months (n = 17) | | | | | | 12% | | | | | | 41% | | | | | | | | | 6% | | 6% |
| | 2015 | 10/13 | 2–71 months (n = 14) | | | | | | | | | | | | 43% | | | | | | | 7% | 7% | | 7% | |
| Iceland [76] | 2012–2017 | 10 | 0–23 months (n = 273) | 0% | 1% | 0% | 1% | 0% | 17% | 2% | | | 0% | 0% | 2% | 6% | 5% | | | | 0% | 4% | | 3% | | |
| | 2012–2017 | 10 | 24–47 months (n = 273) | 0% | 5% | 0% | 2% | 0% | 11% | 9% | | | 0% | 0% | 6% | 10% | 5% | | | | 2% | 1% | | 5% | | |
| Italy [91] | 2015–2016 | 13 | <24 months (n = 23 Vaccinated) | 0% | 0% | 0% | 9% | 0% | 9% | 0% | 17% | 0% | 0% | 0% | 13% | 0% | 0% | 13% | | | | | 0% | | | |
| | 2015–2016 | 13 | <24 months (n = 1 Not vaccinated) | 0% | 0% | 0% | 0% | 0% | 0% | 0% | 0% | 0% | 0% | 0% | 100% | 0% | 0% | 100% | | | | | 0% | | | |
| | | 13 | 24–59 months (n = 13 Vaccinated) | 8% | 0% | 0% | 0% | 0% | 8% | 0% | 14% | 8% | 0% | 0% | 8% | 0% | 0% | 15% | | | | | 0% | | | |
| | | 13 | 24–59 months (n = 1 Not vaccinated) | 0% | 0% | 0% | 0% | 0% | 0% | 0% | 0% | 0% | 0% | 0% | 0% | 0% | 0% | 0% | | | | | 100% | | | |
| Romania [69] | 2009–2014 | None | <60 months (n = 68) | | 12% | | 13% | | 35% | 16% | 78% | | | | | 3% | 7% | | | 90% | | | | | | |
| Slovakia [92] | 2016–2017 | 10/13 | 0–71 months (n = 295) | | | | | | | | | | | | 22% | | 41% | | | | | | | | | |
| Spain [70] | 2009–2012 | 7→ | 3–36 months (n = 24) | | 4% | | 4% | | 13% | | 14% | | | | | | 17% | | 8% | | | | | 4% | | 4% |
| | | 10/13(P) | | | | | | | | | | | | | | | | | 8% | | | | | | | |
| **Nasopharynx** | | | | | | | | | | | | | | | | | | | | | | | | | | | |
| Belgium [71, 78] | 2016 | 13 | 6–30 months (n = 27, | 0% | | | 4% | | 4% | 0% | | 0% | | | 0% | 0% | 0% | | | | | | | 4% | 14% | 7% |
| | 2016 | 13 | n = 79) | 0% | | | 4% | | 4% | 0% | | 0% | | | 0% | 0% | 0% | | | | | | | 4% | 15% | 7% |
| | 2016–2017 | 13 | | 0% | | | 0% | | 3% | 0% | | 0% | | | 3% | 0% | 3% | | | | | | | 5% | 6% | 10% |
| | 2017 | 13 | | | | | 0% | | 2% | 0% | | 0% | | | 5% | 0% | 4% | | | | | | | 5% | 6% | 10% |
| | 2018 | 13 | | | | | 0% | | 3% | 0% | | 0% | | | 2% | 0% | 6% | | | | | | | 6% | 7% | 6% |
| | 2016–2018 | 13 | | | | | | | | | | | | | | | | | | | | 0% | | | |

*(Continued)*

**Table 5.** (Continued)

| Location | Data period | PCVs included during period | Population (n, by order of publication) | PCV7 | | | | | | | | PCV10 | | | PCV13 | | | | | | PCV15 | | PCV20 | | | | |
|---|---|---|---|---|---|---|---|---|---|---|---|---|---|---|---|---|---|---|---|---|---|---|---|---|---|---|---|
| | | | | 4 | 6B | 9V | 14 | 18C | 19F | 23F | PCV7 serotypes | 1 | 5 | 7F | 3 | 6A | 19A | 1, 3, 5, 6A, 7F, 19A | 3, 6A, 19A | PCV13 serotypes | 22F | 33F | 8 | 10A | 11A | 12F | 15B |
| France [12, 63, 72, 73, 75, 93] | 2011–2012 | 13 | 6–24 months (n = 976) | | | | | | | | 3% | | | | | | | | | | | | | | | | |
| | 2010–2013 | 13 | 6–24 months (n = NS) | | | | | | | | 3% | | | | | | 8% | 11% | | | | | | | 9% | | |
| | 2014 | 13 | 6–24 months (n = 7,991) | | | | | | | | 1% | | | | | | 1% | | | | | | | | 7% | | |
| | 2013–2016 | 13 | 6–24 months (n = 1994) | | | | | | | | 3% | | | | | | | | | | | | | | 10% | | |
| | 2011–2018 | 13 | 6–24 months (n = 2409) | | | | | | | | 3% | | | | | | | | | | | | | | | | |
| | 2001–2019 | None→ 7→13 | 6–24 months (n = 10,740) | | | | | | <3% | | | | | | <3% | | <3% | | | | | | | 5–10% | 5–10% | | |
| Switzerland [94] | 2011–2015 | 13 | <12 months (n = 108) | | | | | | | | 10% | | | | | | | 22% | | | | | | | | | |
| | | 13 | 24–59 months (n = 108) | | | | | | | | 9% | | | | | | | 29% | | | | | | | | | |
| **Not specified** | | | | | | | | | | | | | | | | | | | | | | | | | | | |
| France [95] | 2015–2018 | 13 | 6–23 months (n = 1,515) | | | | | | | | | | | | | | | | | 7% | 2% | 2% | 1% | 5% | 10% | 1% | |

Abbreviations (P), only included in NIP for certain regions. NS, not specified

Notes: Distributions reported as 0% represent serotypes that were examined in the studies but were not detected in any sample

Values are rounded with no decimals. Sample sizes vary by study

**Table 6. Serotype distributions over time for reported "non-PCV" serotypes.**

| Record | Country | Age group | Test location | Serotypes | Time period | PCV in NIP | n | % |
|---|---|---|---|---|---|---|---|---|
| Rybak et al 2018 [73] | France | 6 to 24 months | NP | Non-PCV13 and non-6C serotypes | 2001–2002 | No PCV | 867 | 14.30% |
| | | | | | 2003–2005 | 7 (R) | 1229 | 32.40% |
| | | | | | 2006–2010 | 7 | 2488 | 56.10% |
| | | | | | 2011–2012 | 13 | 976 | 83.70% |
| | | | | | ≥2013 | 13 | 1994 | 91.70% |
| Kempf et al 2015 [11] | France | 0–16 years | MEF | 15A | 2001 | No PCV | 152 | 0.30% |
| | | | | 15A | 2011 | 13 | 341 | 6.60% |
| | | | | 23A | 2001 | No PCV | 152 | 0.60% |
| | | | | 23A | 2011 | 13 | 341 | 3.90% |
| Allemann et al 2017 [94] | Switzerland | <1 year | NP | Non-PCV13 isolates | 2004–2006 | 7 (R)→ 7 | 314 | 23.60% |
| | | | | | 2007–2010 | 7 | 188 | 39.40% |
| | | | | | 2011–2015 | 13 | 109 | 67.90% |
| | | 2–4 years | NP | Non-PCV13 isolates | 2004–2006 | 7 (R)→ 7 | 251 | 22.70% |
| | | | | | 2007–2010 | 7 | 145 | 35.20% |
| | | | | | 2011–2015 | 13 | 77 | 62.30% |
| | | <1 year | MEF | Non-PCV13 isolates | 2004–2006 | 7 (R)→ 7 | 17 | 0.00% |
| | | | | | 2007–2010 | 7 | 20 | 25.00% |
| | | | | | 2011–2015 | 13 | 16 | 18.80% |
| | | 2–4 years | MEF | Non-PCV13 isolates | 2004–2006 | 7 (R)→ 7 | 8 | 12.50% |
| | | | | | 2007–2010 | 7 | 16 | 12.50% |
| | | | | | 2011–2015 | 13 | 8 | 25.00% |
| Ouldali et al 2019 [75] | France | 6–24 months | NP | non-PCV13 serotypes | 2001–2002 | No PCV | 719 | 23.10% |
| | | | | | 2003–2010 | 7 (R) → 7 → 13 | 2589 | 50.10% |
| | | | | | 2011–2018 | 13 | 2409 | 89.20% |
| Wouters et al 2019 [71] | Belgium | 6–30 months | NP | 23B | 2016 | 13 | 27 | 11.50% |
| | | | | 23B | 2017 | 13 | 79 | 16.50% |
| | | | | 23A | 2016 | 13 | 27 | 3.80% |
| | | | | 23A | 2017 | 13 | 79 | 3.80% |
| | | | | 21 | 2016 | 13 | 27 | 0.00% |
| | | | | 21 | 2017 | 13 | 79 | 5.10% |
| Quirk et al 2018 [76] | Iceland | 0-<2 years | MEF | 21 | 2009–2011 | 7 (R) → 10 | 401 | 0.00% |
| | | | | 21 | 2012–2017 | 10 | 273 | 3.30% |
| | | | | 38 | 2009–2011 | 7 (R) → 10 | 401 | 0.50% |
| | | | | 38 | 2012–2017 | 10 | 273 | 0.00% |
| | | | | 15A | 2009–2011 | 7 (R) → 10 | 401 | 0.00% |
| | | | | 15A | 2012–2017 | 10 | 273 | 1.50% |
| | | | | 16F | 2009–2011 | 7 (R) → 10 | 401 | 0.20% |
| | | | | 16F | 2012–2017 | 10 | 273 | 0.00% |
| | | | | 19C | 2009–2011 | 7 (R) → 10 | 401 | 0.20% |
| | | | | 19C | 2012–2017 | 10 | 273 | 0.00% |
| | | | | 23A | 2009–2011 | 7 (R) → 10 | 401 | 0.50% |
| | | | | 23A | 2012–2017 | 10 | 273 | 6.60% |
| | | | | 23B | 2009–2011 | 7 (R) → 10 | 401 | 0.20% |
| | | | | 23B | 2012–2017 | 10 | 273 | 3.70% |
| | | | | 35B | 2009–2011 | 7 (R) → 10 | 401 | 0.00% |
| | | | | 35B | 2012–2017 | 10 | 273 | 2.20% |
| | | | | 35F | 2009–2011 | 7 (R) → 10 | 401 | 0.00% |
| | | | | 35F | 2012–2017 | 10 | 273 | 0.70% |
| | | | | 6C | 2009–2011 | 7 (R) → 10 | 401 | 0.20% |
| | | | | 6C | 2012–2017 | 10 | 273 | 12.60% |
| | | | | Other | 2009–2011 | 7 (R) → 10 | 401 | 2.70% |
| | | | | Other | 2012–2017 | 10 | 273 | 14.70% |

*(Continued)*

**Table 6.** (Continued)

| Record | Country | Age group | Test location | Serotypes | Time period | PCV in NIP | n | % |
|--------|---------|-----------|---------------|-----------|-------------|------------|---|---|
| Alonso et al 2013 [79] | Spain | <5 years | MEF | 6C | 1999–2001 | No PCV | 117 | 0.00% |
| | | | | 6C | 2002–2004 | No PCV | 209 | 0.50% |
| | | | | 6C | 2005–2007 | No PCV → 7 (P) | 265 | 1.10% |
| | | | | 6C | 2008–2010 | 7 (P) → 10/13 (P) | 227 | 4.40% |
| | | | | 15A | 1999–2001 | No PCV | 117 | 0.00% |
| | | | | 15A | 2002–2004 | No PCV | 209 | 0.00% |
| | | | | 15A | 2005–2007 | No PCV → 7 (P) | 265 | 1.10% |
| | | | | 15A | 2008–2010 | 7 (P) → 10/13 (P) | 227 | 1.30% |
| | | | | 16F | 1999–2001 | No PCV | 117 | 0.90% |
| | | | | 16F | 2002–2004 | No PCV | 209 | 1.40% |
| | | | | 16F | 2005–2007 | No PCV → 7 (P) | 265 | 2.60% |
| | | | | 16F | 2008–2010 | 7 (P) → 10/13 (P) | 227 | 2.60% |
| | | | | 21 | 1999–2001 | No PCV | 117 | 0.90% |
| | | | | 21 | 2002–2004 | No PCV | 209 | 1.00% |
| | | | | 21 | 2005–2007 | No PCV → 7 (P) | 265 | 3.80% |
| | | | | 21 | 2008–2010 | 7 (P) → 10/13 (P) | 227 | 1.80% |
| | | | | 23A | 1999–2001 | No PCV | 117 | 0.90% |
| | | | | 23A | 2002–2004 | No PCV | 209 | 1.00% |
| | | | | 23A | 2005–2007 | No PCV → 7 (P) | 265 | 1.50% |
| | | | | 23A | 2008–2010 | 7 (P) → 10/13 (P) | 227 | 0.90% |
| | | | | 23B | 1999–2001 | No PCV | 117 | 0.00% |
| | | | | 23B | 2002–2004 | No PCV | 209 | 1.00% |
| | | | | 23B | 2005–2007 | No PCV → 7 (P) | 265 | 1.10% |
| | | | | 23B | 2008–2010 | 7 (P) → 10/13 (P) | 227 | 1.30% |
| | | | | 31 | 1999–2001 | No PCV | 117 | 0.00% |
| | | | | 31 | 2002–2004 | No PCV | 209 | 0.00% |
| | | | | 31 | 2005–2007 | No PCV → 7 (P) | 265 | 0.40% |
| | | | | 31 | 2008–2010 | 7 (P) → 10/13 (P) | 227 | 2.20% |
| | | | | 38 | 1999–2001 | No PCV | 117 | 0.00% |
| | | | | 38 | 2002–2004 | No PCV | 209 | 0.00% |
| | | | | 38 | 2005–2007 | No PCV → 7 (P) | 265 | 0.40% |
| | | | | 38 | 2008–2010 | 7 (P) → 10/13 (P) | 227 | 0.00% |
| | | | | Other | 1999–2001 | No PCV | 117 | 0.00% |
| | | | | Other | 2002–2004 | No PCV | 209 | 4.30% |
| | | | | Other | 2005–2007 | No PCV → 7 (P) | 265 | 3.40% |
| | | | | Other | 2008–2010 | 7 (P) → 10/13 (P) | 227 | 2.60% |
| | | | | Non-typable | 1999–2001 | No PCV | 117 | 2.60% |
| | | | | Non-typable | 2002–2004 | No PCV | 209 | 1.00% |
| | | | | Non-typable | 2005–2007 | No PCV → 7 (P) | 265 | 1.50% |
| | | | | Non-typable | 2008–2010 | 7 (P) → 10/13 (P) | 227 | 0.00% |

Different disease-specific questionnaires have been used to assess quality of life in AOM. The Otitis Media-6 (OM-6) questionnaire was used in three studies conducted in Denmark. The questionnaire consists of six domains of functional health status (FHS) to measure the child's physical suffering, hearing loss, speech impairment, emotional distress, activity limitations and the caregiver's concern, as well as a numerical rating scale (NRS-child) to assess the child's QoL [104]. Parents answer the questionnaire based on symptoms experienced by the child in the past four weeks and a summary score can be obtained by adjusting the scores to a scale of 0 to 100 [104–106]. Additionally, the Caregiver Impact Questionnaire was used in the study by Heidemann et al 2014 [107]. Two studies used self-developed QoL questionnaires [108, 109].

## Discussion

This systematic literature review is, to our knowledge, the first study that has gathered data from published literature on AOM incidence, etiology, antibiotic resistance and serotype distribution for *S. pneumoniae*, costs of illness and impact on quality of life across Europe. The identified data provided evidence of a shift in residual burden of AOM after the introduction of PCVs for countries where data was available. This was notable on serotype distribution and antibiotic resistance for non-vaccine serotypes. There was also evidence of a decrease in hospitalization rates due to AOM and a general decrease in antibiotic resistance rates in *S. pneumoniae*, which are contributing factors especially for the economic burden of AOM. However, due to the limited availability of data and the methodological differences of the included literature, the authors advise caution in applying the conclusions across the European region as a whole.

The identified records revealed a wide variation in estimates between countries and over time for all outcomes assessed. Other studies have also acknowledged that the variability of AOM data limited comparison across studies [4, 35]. This data heterogeneity is likely caused by many factors. For instance, the authors encountered 18 different definitions for AOM used in the included studies (S7 Table), which could contribute to the high variability in incidence rates between publications. In addition, the differences in the ages of study populations also hindered a more meaningful comparison of incidence, etiology and serotype distribution estimates between and within the included countries.

Furthermore, differences in clinical practice might affect comparability across studies, such as the choice of sampling and analytical methods in the analysis for pathogen etiology and pneumococcus serotype distribution of AOM. For example, the use of polymerase chain reaction (PCR) versus bacterial culture for the detection of *S. pneumoniae* can impact the accuracy of estimates [110]. As mentioned in the Etiology section, estimates from countries with data from the same years and target populations, but different sampling methods, varied considerably. This finding might indicate that other factors could explain variation between samples collected from MEF and NP swabs. Based on previous literature, the correlation between bacterial cultures in MEF and the nasopharynx is strong but not absolute, making it difficult to assume the different methods would produce similar results [69, 111–115].

Variations in the aggregation of data also complicated the prospect of comparisons. For example, previous studies often reported the distribution of non-PCV serotypes as a group, without specifying which individual serotypes had been identified. This imposed a challenge, particularly when analyzing older studies, since the group of non-PCV serotypes at that time likely contained some serotypes that were later incorporated into the most recent PCVs [85, 116]. However, since they were not reported individually, this added to the difficulty of identifying trends over time for the grouped serotypes, particularly for the years between PCV7 and PCV10/13 introduction.

A similar complication was also noticed for the economic burden estimates; the types of costs included in the major cost categories varied widely, as did the methods of calculation. This was particularly evident with the reporting of direct medical costs and estimation of cost per AOM episode shown in Fig 6, since the original publication that reported the remarkably low cost per AOM episode in the UK did not specify which costs were considered in their calculations [100]. Moreover, there were variations in non-medical costs (e.g., inclusion of public transportation, estimation of time spent vs. distance travelled, etc.) and indirect costs (e.g., productivity loss vs. caregiver burden, the inputs chosen to estimate those, etc.). The differences in methods, combined with the other factors discussed above, precluded any sort of meaningful comparison between countries or across time in these cost categories.

In addition to the data variability, there were large data gaps in both clinical and economic outcomes for some European countries and certain time-periods; only 11 of the 31 included countries had at least one record reporting on every outcome of interest. For six countries, there were no studies with relevant data identified at all. Furthermore, though the literature searches were limited to the last ten years, almost 30% of the data identified came from before 2011.

Despite these complications, the authors were able to identify a few trends in the data. First, there were indications of a general reduction in hospitalization rate after the introduction of PCVs. This factor should point to a reduction in the economic burden of AOM over time, as a decrease in the occurrence or the average severity and duration of AOM episodes would mean a decrease in the resources and time required to treat those patients. However, time trends in the economic burden estimates that were identified were limited, making confirmation of this hypothesis difficult. Second, a reduction in antibiotic-resistant pneumococci was also identified in the literature, which is line with previous studies [117, 118]. On the other hand, there was evidence of an increase in non-vaccine serotypes over time for countries that had introduced PCVs into their NIP. This serotype replacement phenomenon has already been described for other manifestations of pneumococcal disease. Also, there was limited evidence of an increase in antibiotic resistance within non-vaccine serotypes. Both of these trends suggest that a residual burden remains despite the shift towards higher-valency PCVs.

The natural assumption for addressing residual burden would be to increase the valency of available PCVs. However, recent studies have shown that 'more is not necessarily better'; a systematic review by Mungall et al in 2022 identified vaccine failures and breakthrough IPD with PCV10 and PCV13 in children up to 5 years and warned of the importance of addressing incomplete protection against certain serotypes [119]. Also, a 2020 study by Løchen et al identified a reduction in the marginal benefit of further increasing the valency of vaccines, as the vanishing of a dominant serotype after PCV introduction mitigates the benefits of targeting and covering additional serotypes [120].

This review had certain limitations when it came to methods and data synthesis. Regarding the former, the exclusion of grey literature, such as surveillance reports or policy documents, from the scope of this review limited the potential for relevant data in some of the outcome categories. However, as this was a review of data available in published literature since 2011, the data gaps are part of the narrative of this review. Additionally, given that AOM was the main focus and also a required search term, many relevant articles concerning PCV-coverage were not captured since these vaccines are also protective against other pneumococcal disease manifestations more well captured in terms of surveillance. Also, records regarding PCV-coverage were still included in the summary statistics as they were indeed part of the original search and extraction process, but there was no synthesis done based on the reported estimates due to the clear risk of bias arising from the factors described above. Finally, additional indicators for a change in AOM burden, such as the rates of physician office visits and antibiotic prescriptions, were not explored in this study, but could have provided further insight.

## Conclusions

This study compiled the most recent published evidence on the clinical and economic burden of AOM among European children. Despite the data gaps, it was possible to obtain comparable estimates for reduction in both hospitalization rates and antibiotic resistance of *S. pneumoniae*, as well as evidence of an increase in residual burden for pneumococcal AOM from non-vaccine serotypes, in the countries with available data. The lack of a homogenous trend in incidence rates across countries and the increased frequency of new serotypes indicate that the

burden of pneumococcal AOM is still meaningful, despite the introduction of PCVs. Thus, vaccination programs should be maintained with high coverage rates and re-evaluated when new PCVs become available.

The data variability identified in the present study confirms the need for more standardization when reporting information related to AOM. For instance, future researchers could use the STROBE guidelines for reporting cross-sectional studies when reporting data on AOM [121]. Also, when applicable, the World Health Organization's recommendations for detecting carriage of *S. pneumoniae* in NP samples should be followed [122]. Another way to address this issue is with the implementation of surveillance systems to monitor VCR, serotype distribution, etiology, antibiotic resistance, and disease burden. Surveillance systems are instrumental for understanding the burden of pneumococcal diseases, including AOM, which can guide vaccination programs. Even though surveillance systems for invasive pneumococcal diseases exist in all of the European countries included in this study, there is limited information on surveillance of noninvasive forms of pneumococcal disease, like AOM [123]. Thus, it is still difficult to provide comprehensive, accurate and up-to-date estimates of AOM-burden and epidemiology for the European region using published literature. This study confirms that there are still significant data gaps within the published literature, which points to a greater need for surveillance systems monitoring AOM, while also providing some evidence of a reduction in the burden of AOM over time in countries with available data, likely attributed to the introduction of PCVs.

## Supporting information

**S1 Fig. Introduction status of PCVs in European countries over time.** Notes: Ⓡ: Vaccine is not administered to the entire population, only to specific risk groups. (P): Vaccine is administered only in certain regions of the country. Sources: [9, 19–28].
(TIF)

**S1 Table. PubMedⓇ search terms.**
(DOCX)

**S2 Table. Type of data extracted from included records.**
(DOCX)

**S3 Table. List of included records.**
(DOCX)

**S4 Table. Number of included records within each category, divided by country.**
(DOCX)

**S5 Table. Records including antibiotic testing of S. pneumoniae in general and of specific serotypes.**
(DOCX)

**S6 Table. Summary data of the records that reported on the economic burden of AOM.**
(DOCX)

**S7 Table. Definition groups for AOM for the included studies.**
(DOCX)

**S8 Table. PRISMA checklist.**
(DOCX)

**S1 File. Data extraction grid.**
(XLSX)

## Author Contributions

**Conceptualization:** Heloisa Ricci Conesa, Eleana Tsoumani.

**Data curation:** Heloisa Ricci Conesa, Helena Skröder.

**Formal analysis:** Heloisa Ricci Conesa, Helena Skröder, Nicholas Norton.

**Funding acquisition:** Goran Bencina, Eleana Tsoumani.

**Methodology:** Helena Skröder.

**Project administration:** Nicholas Norton.

**Resources:** Eleana Tsoumani.

**Supervision:** Nicholas Norton, Goran Bencina, Eleana Tsoumani.

**Validation:** Heloisa Ricci Conesa, Helena Skröder, Nicholas Norton, Goran Bencina, Eleana Tsoumani.

**Writing – original draft:** Heloisa Ricci Conesa, Helena Skröder, Nicholas Norton.

**Writing – review & editing:** Heloisa Ricci Conesa, Helena Skröder, Nicholas Norton, Goran Bencina, Eleana Tsoumani.

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
