## [Decision Letter · Decision Letter 0]

22 Nov 2022

PONE-D-22-29637Clinical and economic burden of acute otitis media caused by *Streptococcus pneumoniae* in European children – a systematic literature review of published evidencePLOS ONE

Dear Dr. Conesa,

Thank you for submitting your manuscript to PLOS ONE. After careful consideration, we feel that it has merit but does not fully meet PLOS ONE’s publication criteria as it currently stands. Therefore, we invite you to submit a revised version of the manuscript that addresses the points raised during the review process.

We look forward to receiving your revised manuscript.

Kind regards,

Sethu Thakachy Subha, M.S

Academic Editor

PLOS ONE

**Journal Requirements**:

3. Please expand the acronym “MSD” (as indicated in your financial disclosure) so that it states the name of your funders in full.

"This study was funded by MSD. ET reviewed the study design. ET and GB reviewed the data collection and analysis, as well as the decision to publish. GB reviewed the preparation of the manuscript."

"I have read the journal's policy and the authors of this manuscript have the following competing interests: ET and GB are employees of MSD. HRC, HS and NN are employees of Quantify Research, which received consulting fees from MSD to support the preparation and development of the manuscript. The authors report no other conflicts of interest in this work."

We note that one or more of the authors are employed by a commercial company: MSD, Quantify Research

(2) Please also provide an updated Competing Interests Statement declaring this commercial affiliation along with any other relevant declarations relating to employment, consultancy, patents, products in development, or marketed products, etc.  

Within your Competing Interests Statement, please confirm that this commercial affiliation does not alter your adherence to all PLOS ONE policies on sharing data and materials by including the following statement: ""This does not alter our adherence to  PLOS ONE policies on sharing data and materials.” (as detailed online in our guide for authors http://journals.plos.org/plosone/s/competing-interests). 

If this adherence statement is not accurate and  there are restrictions on sharing of data and/or materials, please state these. Please note that we cannot proceed with consideration of your article until this information has been declared.

6. We note that you have indicated that data from this study are available upon request. PLOS only allows data to be available upon request if there are legal or ethical restrictions on sharing data publicly. For more information on unacceptable data access restrictions, please see http://journals.plos.org/plosone/s/data-availability#loc-unacceptable-data-access-restrictions. 

7.  We note that Figure 4 in your submission contain map images which may be copyrighted. All PLOS content is published under the Creative Commons Attribution License (CC BY 4.0), which means that the manuscript, images, and Supporting Information files will be freely available online, and any third party is permitted to access, download, copy, distribute, and use these materials in any way, even commercially, with proper attribution. For these reasons, we cannot publish previously copyrighted maps or satellite images created using proprietary data, such as Google software (Google Maps, Street View, and Earth). For more information, see our copyright guidelines: http://journals.plos.org/plosone/s/licenses-and-copyright.

(1) You may seek permission from the original copyright holder of Figure 4 to publish the content specifically under the CC BY 4.0 license.  

8. Please upload a new copy of Figure S1 as the detail is not clear. Please follow the link for more information:

https://blogs.plos.org/plos/2019/06/looking-good-tips-for-creating-your-plos-figures-graphics/

https://blogs.plos.org/plos/2019/06/looking-good-tips-for-creating-your-plos-figures-graphics/

9. We note that this manuscript is a systematic review or meta-analysis; our author guidelines therefore require that you use PRISMA guidance to help improve reporting quality of this type of study. Please upload copies of the completed PRISMA checklist as Supporting Information with a file name “PRISMA checklist”.

Reviewers' comments:

Reviewer's Responses to Questions

**Comments to the Author**

1. Is the manuscript technically sound, and do the data support the conclusions?

Reviewer #1: Partly

Reviewer #2: Yes

2. Has the statistical analysis been performed appropriately and rigorously? 

Reviewer #1: I Don't Know

Reviewer #2: N/A

3. Have the authors made all data underlying the findings in their manuscript fully available?

Reviewer #1: No

Reviewer #2: Yes

4. Is the manuscript presented in an intelligible fashion and written in standard English?

Reviewer #1: Yes

Reviewer #2: Yes

5. Review Comments to the Author

Reviewer #1: AOM burden is quantified by the number of AOM cases/recurrent AOM cases, and not by the number of AOM hospitalizations or antibiotic resistance patterns of bacteria isolated from AOM cases, which becomes more and more rare since myringotomy is not routinely performed anymore.

I think that he authors worked pretty hard on this manuscript, and I will strongly encourage them to revise it.

The abstract should be clearer: which PCVs are included in this review? the results given- to which vaccine? from 2011 there are many good quality papers that cover PCV7, PHiD-CV (PCV10) and PCV13. This distinction should be clearly stated and referred to. What were your criteria and outcomes in your research? The conclusions part is very general and not based on your results—when you say that AOM burden in Europe is still very high—you did not give any numbers in the previous Results paragraph.

Introduction: Give PHiD-CV and PCV13 timelines. PCV15/PCV20—mention them in the Discussion section. Redefine your research question, it still remains unclear.

Methods can be shortened. Leave only what's important. How did you actually assess the change in AOM burden after PCVs were available?

Results. It is essential to state for each paper that you show in Table 2 what was the immunization status of the children-vaccinated (with what?), unvaccinated? so we can understand the figures. Do you have any pre-vaccine data for comparison purposes?. It appears that you missed many papers from France. Here are a few of them that may be useful:

- Bacterial causes of otitis media with spontaneous perforation of the tympanic membrane in the era of the 13-valent pneumococcal conjugate vaccine.

Levy C, Varon E, Ouldali N, Wollner A, Thollot F, Corrard F, Werner A, Béchet S, Bonacorsi S, Cohen R.

PLoS One. 2019 Feb 1;14(2):e0211712. doi: 10.1371/journal.pone.0211712.

- The multifaceted impact of pneumococcal conjugate vaccine implementation in children in France between 2001 to 2014.

Cohen R, Biscardi S, Levy C.

Hum Vaccin Immunother. 2016;12(2):277-84. doi: 10.1080/21645515.2015.1116654.

- Community antibiotic prescribing for children in France from 2015 to 2017: a cross-sectional national study.

Trinh NTH, Cohen R, Lemaitre M, Chahwakilian P, Coulthard G, Bruckner TA, Milic D, Levy C, Chalumeau M, Cohen JF.

J Antimicrob Chemother. 2020 Aug 1;75(8):2344-2352. doi: 10.1093/jac/dkaa162.

- Antibiotic Resistance of Potential Otopathogens Isolated From Nasopharyngeal Flora of Children With Acute Otitis Media Before, During and After Pneumococcal Conjugate Vaccines Implementation.

Rybak A, Levy C, Bonacorsi S, Béchet S, Vié le Sage F, Elbez A, Varon E, Cohen R.

Pediatr Infect Dis J. 2018 Mar;37(3):e72-e78. doi: 10.1097/INF.0000000000001862.

-

By generally describing that "PCVs" reduced AOM, it is hard to figure out which one of them was a key player.

I highly recommend the authors to look at other non-European studies where there is a clear differentiation between each PCV. This truly determines the epidemiology and bacteriology of AOM nowadays. Resistance of pneumococci- again, it is mandatory to state whether there were bacteria grown in ear cultures or from nasopharyngeal swabs. as the concordance rate between the nasopharynx and the middle ear is only moderate...

Discussion is nicely written, but the Conclusion parts is too general .

Reviewer #2: This review, titled “Clinical and economic burden of acute otitis media caused by Streptococcus pneumoniae in European children – a systematic literature review of published evidence” aimed to provide a current, comprehensive overview of AOM burden in Europe including economic burden, data on incidence, etiology, pneumococcal serotype and antibiotic resistance. The analyses provide trends in these outcomes over time (published since 2011) including PCV coverage and switching.

The authors are to be commended on a massive undertaking and well-presented summary of findings from 107 records detailed in 5 Tables, 7 Figures and 8 supporting documents.

I have made minor suggestions for the text, and somewhat greater changes suggested for some tables, to improve reader navigation of the data presented.

Further recommendations

The authors should consider making specific or explicit recommendations for future researchers reporting on this topic, such as the STROBE guidelines for (minimal data) reporting cross-sectional studies. (https://www.strobe-statement.org/index.php?id=available-checklists) see Table 8 in https://doi.org/10.1016/j.ijporl.2019.109857 as a suggested format for recommended reporting on otitis media.

For microbiological studies of S. pneumoniae we recommend evaluation of methods against WHO recommendations (see comments for Table 4 and https://doi.org/10.1016/j.vaccine.2013.08.062) see also Table 4 in https://doi.org/10.1016/j.ijporl.2019.109857

Abstract

There is no mention of etiology, serotype shift or economic burden. To allow these findings to be included in the abstract word limit, lines 35 to 37 could be condensed. For instance, change to ‘.. wide variation in study methodology and reporting, which limited comparisons of all outcomes.’ Then delete along with data gap and delete sentence on L36-37 ‘Comparisons …methods.’

Line 39 after “depending on country” add “, PCV type and time since PCV introduction”)

Line 44 delete due to variation in the methodologies and finish with ‘improved by standardised methodology, reporting and wider use of surveillance systems’

Introduction

L54-55 change to “… caused by vaccine serotype S. pneumoniae.”

L60 change have to has

L62 change new to non-vaccine (the serotypes are not ‘new’, non-vaccine serotypes emerge as the vaccine serotypes are eliminated)

L67-68 suggest change wording after ; to ;’ in addition to the serotypes present in PCV13, PCV15 contains 22F and 33F, and PCV20 contains serotypes 8, 10A, 11A, 12F, and 15B’

L84 remove comma after collecting

L85 add ‘of age’ after < 5 years

Methods

L92 I could not access the PROSPERO hyperlink in Chrome or Edge

L112-115 Why were abstracts from ISPPD International Symposium on Pneumococci and Pneumococcal Diseases not included as highly relevant.

L125 add reference to your PRISMA flowdiagram Figure 1 – although I have noted that this is illegible in the PDF.

L145 change ‘this type of studies’ to ‘this type of study’

L164 change ‘data was’ to ‘data were’ (data are plural)

Results

L185 Figure 1 PRISMA flowchart is illegible

L191 change ‘…category, 19 of them…’ to ‘…category, only 19 of them…’ (as this sentence commenced with ‘Even though..)

L196 could delete ‘and onwards’

L201 Start with ‘Of the 61 studies ..’ (i.e. delete ‘Out’)

L217 change ‘incidence different’ to ‘incidence were different’

L221 Delete ‘However’ and start sentence with Data

L226 to 230 – include in this section the different vaccine eras and time since PCV introduced or switched.

L237 provide range in proportion positive for both MEF and NP. Change to ‘… ranged from 5.8% to 70% in MEF samples .. in Germany and Romania, respectively and from 10.5% to 83% in nasopharyngeal (NP) samples … in Sweden and Portugal, respectively’

L246 Please elaborate on differences in sampling methods and influence on country-specific data for MEF and NP.

L248 – section on antibiotic resistance – should be reported for MEF and for NP samples as resistance can be quite different according to site (as also noted in section on serotype distribution)

L261 Delete ‘Again’ and structure sentence to read the same as first sentence (L258). Ie “There was a lower level of serotype-specific non-susceptibility in the post-PCV time periods”

L264 Changing Figure S1 as suggested below will help understand mid-PCV period and years of PCV switch

L283 It is important to recognise and distinguish between date of publication and date of data collection (as has been done elsewhere).

L284 change ‘contained data from 2011’ to ‘contained data published from 2011’

Line 426 Funding. It is noteworthy that this review was commissioned and conducted by Merck Sharp & Dome who are developing expanded valency pneumococcal conjugate vaccines. Please include role of funders in design, analysis, decision to publish. Author contributions on page 1. It appears that MSD conceptualized and developed design, and had input into extraction, synthesis and writing of the manuscript. Please clarify.

TABLES and FIGURES

Figure 4

is illegible

Figure 5

add N to each bar, proportions alone are not very informative. X-axis is illegible.

Figures 6 and 7

These two Figures could be combined which would show the country differences in proportion of total cost that is direct cost.

Table 1

For conference abstracts add final date

Table 2

Column ‘Time interval’ – add PCV type, year of introduction or switch (e.g. 2011. PCV10. 2001). Perhaps change column title from ‘Time interval’ to ‘Period of data collection’

Table 3

Consider rounding decimal places. Add which PCV in Columns pre-PCV (or no PCV) Post-PCV (Iceland used PCV10). Reference 36 title states “transition era …PCV7 to PCV13.., 2010 to 2013”. Are the data presented in the table pre- and post-, excluding transition era?

Table 4

Column Time interval – add PCV (PCV10 or PCV13), and year of PCV introduction (e.g. 2011. PCV10. 2001). Perhaps change column title from ‘Time interval’ to ‘Period of data collection’

Sampling method can be the major explanation for differences in detection of pneumococci – consider adding a column for ‘Sampling Method’. This could be culture vs PCR, or compliance with WHO standard method (O'Brien,K.L and Nohynek,H. 2003. Report from a WHO working group: standard method for detecting upper respiratory carriage of Streptococcus pneumoniae OR Satzke, C. et al. 2013 Standard method for detecting upper respiratory carriage of Streptococcus pneumoniae: Updated recommendations from the World Health Organization Pneumococcal Carriage Working Group) 10.1016/j.vaccine.2013.08.062.)

95% Confidence Intervals for the point estimates would be helpful, given huge differences in sample size. Column Estimate (%, 95%CI)

Table 5

Column Time interval – add PCV (PCV10 or PCV13), and year of PCV introduction (e.g. 2011. PCV10. 2001). Perhaps change column title from ‘Time interval’ to ‘Period of data collection’

Column Population – add N for each age range and year (e.g. 0-23 months. N)

The row totals for PCV7 serotypes, PCV13 serotypes are incomplete, and would be helpful to include.

Add a final column for row total of non-PCV serotypes (i.e. 100 — (PCV13+PCV15 +PCV20))

SUPPORTING DOCUMENTS

Table S3

As year of publication is given in Column 1, change column ‘Year of publication’ to ‘Year of data collection’

Table S5

Column S. pneumoniae could you split this into detection method (culture, PCR) and serotype (Y/N)

Table S7

the grey highlights are not visible

Figure S1

Instead of yes, I suggest inserting PCV type i.e. PCV7, PCV10 or PCV13. [this will help track country differences in PCV and switch between PCVs – as noted L211 to 223, page 11 and also L265-266, page 15]

Further recommendations

The authors should consider making specific or explicit recommendations for future researchers reporting on this topic, such as the STROBE guidelines for (minimal data) reporting cross-sectional studies. (https://www.strobe-statement.org/index.php?id=available-checklists) see Table 8 in https://doi.org/10.1016/j.ijporl.2019.109857 as a suggested format for recommended reporting on otitis media.

For microbiological studies of S. pneumoniae we recommend evaluation of methods against WHO recommendations (see comments for Table 4 and https://doi.org/10.1016/j.vaccine.2013.08.062) see also Table 4 in https://doi.org/10.1016/j.ijporl.2019.109857

6. PLOS authors have the option to publish the peer review history of their article (what does this mean?). If published, this will include your full peer review and any attached files.

Reviewer #1: **Yes: **Tal Marom

Reviewer #2: **Yes: **Professor Amanda Jane Leach AM

---

## [Author Response · Author response to Decision Letter 0]

7 Mar 2023

Dear Review team,

We want to sincerely thank you for providing such an in-depth review of our work. The comments you provided included valid points, which, after we addressed them, have helped improve the transparency and cohesiveness of the manuscript. We hope that you find the revisions we made satisfactory in addressing your questions and concerns, and appreciate the time and effort you've put into this review. 

Thank you very much!

---

## [Decision Letter · Decision Letter 1]

11 Apr 2023

PONE-D-22-29637R1Clinical and economic burden of acute otitis media caused by *Streptococcus pneumoniae* in European children, after widespread use of PCVs – a systematic literature review of published evidencePLOS ONE

Dear Dr. Norton,

Thank you for submitting your manuscript to PLOS ONE. After careful consideration, we feel that it has merit but does not fully meet PLOS ONE’s publication criteria as it currently stands. Therefore, we invite you to submit a revised version of the manuscript that addresses the points raised during the review process.

We look forward to receiving your revised manuscript.

Kind regards,

Sethu Thakachy Subha, M.S

Academic Editor

PLOS ONE

Journal Requirements:

Reviewers' comments:

Reviewer's Responses to Questions

**Comments to the Author**

1. If the authors have adequately addressed your comments raised in a previous round of review and you feel that this manuscript is now acceptable for publication, you may indicate that here to bypass the “Comments to the Author” section, enter your conflict of interest statement in the “Confidential to Editor” section, and submit your "Accept" recommendation.

Reviewer #1: (No Response)

Reviewer #3: (No Response)

2. Is the manuscript technically sound, and do the data support the conclusions?

Reviewer #1: Yes

Reviewer #3: Yes

3. Has the statistical analysis been performed appropriately and rigorously? 

Reviewer #1: Yes

Reviewer #3: Yes

4. Have the authors made all data underlying the findings in their manuscript fully available?

Reviewer #1: Yes

Reviewer #3: Yes

5. Is the manuscript presented in an intelligible fashion and written in standard English?

Reviewer #1: Yes

Reviewer #3: Yes

6. Review Comments to the Author

Reviewer #1: I can't see your corrections. Please outline them and refer to the comments you have received by the reviewers. Thank you

Reviewer #3: This study is very important that it summarized the most recent AOM burden in EU countries comprehensively. I only have a few minor comments for the author to consider:

Line 78-83: what is the age range for this incidence rate? And spell out xx cases per 1000 person years. And which year are the incidence rates in this paragraph for? Is it pre or post PCV introduction?

Line 81-82: Could you clarify this? Is the incidence rate higher in the <1 or 1-4 years old?

Table 1 and other appropriate sections related with the name "economic burden": QoL is humanistic burden, suggest expanding the name to accurately capture the outcomes that this study assessed

Line 248: Is this inpatient stay? Compared to inpatient stay, AOM is more likely to be associated with outpatient visits and the HCRU associated with outpatient/physician office visits is a greater burden to the health care system. Suggest look into the literature on physician office visits, or provide explanations or plan to look into this in the future.

Line 263-264: suggest putting some context around those percentages, are those from pre or post PCV adoption period for those countries?

Line 270: any possible explanations for the difference? Different regions, methods for taking samples, testing method difference?

Line 396: would be interesting to look at other contributing factors such as physician office visits, and antibiotic prescriptions in addition to the antibiotic resistance

7. PLOS authors have the option to publish the peer review history of their article (what does this mean?). If published, this will include your full peer review and any attached files.

Reviewer #1: **Yes: **Sorry, but I need to see the changes made by the authors according to the comments they have received. Please ask them to submit a version with all the changes high lightened. Thank you.

Reviewer #3: No

---

## [Author Response · Author response to Decision Letter 1]

28 Apr 2023

Dear review team,

Thank you again for taking the time to review our study and assess the work so thoroughly; we appreciate your attention to detail and hope that with the revisions we've made to address your valid concerns, we have improved the manuscript to the standards of this journal. 

Per reviewer 1's request, the new track-changes version of the manuscript contains highlights in all sections where we made changes for the second round of revisions, along with comments explaining those changes, and text in the response document to accompany them. If reviewer 1 is also interested in a copy of the first round of revisions with highlighted changes, we would be happy to provide this separately from the changes in this second round, to avoid any confusion regarding versioning.

We hope that the changes meet your expectations and if there is any further clarification needed, please let us know. 

Kind regards,

Authors

---

## [Decision Letter · Decision Letter 2]

27 Dec 2023

Clinical and economic burden of acute otitis media caused by *Streptococcus pneumoniae* in European children, after widespread use of PCVs – a systematic literature review of published evidence

PONE-D-22-29637R2

Dear Dr. Norton,

We’re pleased to inform you that your manuscript has been judged scientifically suitable for publication and will be formally accepted for publication once it meets all outstanding technical requirements.

Kind regards,

Sethu Thakachy Subha, M.S

Academic Editor

PLOS ONE

Additional Editor Comments (optional):

Reviewers' comments:

Reviewer's Responses to Questions

**Comments to the Author**

1. If the authors have adequately addressed your comments raised in a previous round of review and you feel that this manuscript is now acceptable for publication, you may indicate that here to bypass the “Comments to the Author” section, enter your conflict of interest statement in the “Confidential to Editor” section, and submit your "Accept" recommendation.

Reviewer #4: All comments have been addressed

2. Is the manuscript technically sound, and do the data support the conclusions?

Reviewer #4: Yes

3. Has the statistical analysis been performed appropriately and rigorously? 

Reviewer #4: Yes

4. Have the authors made all data underlying the findings in their manuscript fully available?

Reviewer #4: Yes

5. Is the manuscript presented in an intelligible fashion and written in standard English?

Reviewer #4: Yes

6. Review Comments to the Author

Reviewer #4: This review provides valuable insights into the epidemiology of AOM in Europe but there is a large data gaps and lack of quantitative analysis.

7. PLOS authors have the option to publish the peer review history of their article (what does this mean?). If published, this will include your full peer review and any attached files.

Reviewer #4: No

---

## [Editor Report · Acceptance letter]

19 Mar 2024

PONE-D-22-29637R2 

PLOS ONE

Dear Dr. Norton, 

I'm pleased to inform you that your manuscript has been deemed suitable for publication in PLOS ONE. Congratulations! Your manuscript is now being handed over to our production team.

Kind regards, 

on behalf of

Dr. Sethu Thakachy Subha 

Academic Editor

PLOS ONE